# Linker Length Drives Heterogeneity of Multivalent Complexes of Hub Protein LC8 and Transcription Factor ASCIZ

**DOI:** 10.3390/biom13030404

**Published:** 2023-02-21

**Authors:** Douglas R. Walker, Kayla A. Jara, Amber D. Rolland, Coban Brooks, Wendy Hare, Andrew K. Swansiger, Patrick N. Reardon, James S. Prell, Elisar J. Barbar

**Affiliations:** 1Department of Biochemistry and Biophysics, Oregon State University, Corvallis, OR 97331, USA; 2Department of Chemistry and Biochemistry, University of Oregon, Eugene, OR 97403, USA; 3Institute of Molecular Biology, University of Oregon, Eugene, OR 97403, USA; 4NMR Facility, Oregon State University, Corvallis, OR 97331, USA; 5Materials Science Institute, University of Oregon, Eugene, OR 97403, USA

**Keywords:** hub proteins, intrinsic disorder, multivalency, transcription factor, linker length, heterogeneity, dimers, duplexes

## Abstract

LC8, a ubiquitous and highly conserved hub protein, binds over 100 proteins involved in numerous cellular functions, including cell death, signaling, tumor suppression, and viral infection. LC8 binds intrinsically disordered proteins (IDPs), and although several of these contain multiple LC8 binding motifs, the effects of multivalency on complex formation are unclear. Drosophila ASCIZ has seven motifs that vary in sequence and inter-motif linker lengths, especially within subdomain QT2–4 containing the second, third, and fourth LC8 motifs. Using isothermal-titration calorimetry, analytical-ultracentrifugation, and native mass-spectrometry of QT2–4 variants, with methodically deactivated motifs, we show that inter-motif spacing and specific motif sequences combine to control binding affinity and compositional heterogeneity of multivalent duplexes. A short linker separating strong and weak motifs results in stable duplexes but forms off-register structures at high LC8 concentrations. Contrastingly, long linkers engender lower cooperativity and heterogeneous complexation at low LC8 concentrations. Accordingly, two-mers, rather than the expected three-mers, dominate negative-stain electron-microscopy images of QT2–4. Comparing variants containing weak-strong and strong-strong motif combinations demonstrates sequence also regulates IDP/LC8 assembly. The observed trends persist for trivalent ASCIZ subdomains: QT2–4, with long and short linkers, forms heterogeneous complexes, whereas QT4–6, with similar mid-length linkers, forms homogeneous complexes. Implications of linker length variations for function are discussed.

## 1. Introduction

Hub proteins, which interact with many different proteins in an organism, gained recognition around the turn of the century as highly important and often essential parts of an organism’s proteome [1]. Jeong et al. showed in Saccharomyces cerevisiae that 0.7% of proteins interact with 15 or more other proteins, but a single deletion in 62% of these proved to be lethal, three times more than for proteins with a small number of protein partners. Hub proteins can be subdivided according to their structure and their clients. Intrinsic disorder plays a major role in enabling flexibility and promiscuity in hub binding [2]. Thus, hub proteins can be organized into three broad classes: (1) completely disordered interacting with ordered proteins, (2) partially disordered interacting with ordered proteins, and (3) fully ordered interacting with intrinsically disordered proteins. The third class of hub proteins often induces folding of a short linear binding motif in their partner proteins upon binding. Proteins that fit into this class of hub protein include calmodulin, actin, Cdk2, 14-3-3 [2], RCD1-RST [3], Keap1 [4], and LC8 [5]. Due to the structural flexibility of the binding groove of class three hubs and the variability allowed in the partner sequence, they tend to interact with a greater number and wider variety of proteins than those in other classes.

Recognition of the prevalence of intrinsically disordered proteins and protein regions (IDPs/IDRs) and their roles as biologically active proteins has rapidly grown [6]. IDPs and IDRs are characterized by low sequence diversity, lack of hydrophobic residues, abundance of charged residues, and areas of sequence repeats. Due in part to their high number of charged residues, as well as their abundance of short linear binding motifs, disordered regions are promiscuous in their binding interactions and facilitate the formation of many complex large protein assemblies [7]. IDPs/IDRs are also extremely functionally diverse, and in addition to their structural plasticity and dynamic conformational flexibility, they often interact with their binding partners multivalently.

Because they interact with IDPs, it is not uncommon that class three hubs will interact with their partners multivalently. An actin filament in red cell membranes will bind to between five and seven 4.1R proteins [8]. Keap1 binds at two locations on NRF2 to facilitate ubiquitination [4]. The C-terminal domain of calmodulin binds melittin in the absence of Ca^2+^, but upon the addition of Ca^2+^, the N-terminal domain also binds [9]. LC8 binds many of its protein partners multivalently (Figure 1A); however, it is unique in its large number of both multivalent partners and binding motifs on a single protein; for instance, ASCIZ contains 11–16 LC8 recognition sites [10] (Figure 1C and Figure 2A). Compared to monovalent interactions, in which ligands bind a partner at a single site, multivalent interactions involve linked associations of ligands binding to multiple sites [11,12,13]. Multivalent IDP assemblies are considered to belong to one of the following groups: binary complexes, IDP single-chain scaffolds, IDP duplex scaffolds, higher-order IDP associations, and IDP multi-site collective binding ligands [13]. LC8, the focus of this work, folds as a homodimer and assembles IDPs into duplex scaffolds which are composed of two IDP chains connected by one or more bivalent partners with two symmetrical binding sites and/or by self-association interactions within the chain [12,13]. Cases in which the same dimeric ligand binds multiple sites across disordered chains are common for partners of LC8 (Figure 1) [13,14,15].

Within the LC8 homodimer, two symmetrical binding grooves allow LC8 to duplex its intrinsically disordered binding partners (Figure 1A) [11,13,30]. LC8 is an essential protein in animal proteomes [31,32,33] and is confirmed to partner with more than 100 different client IDPs; among these are IDPs performing functions such as intracellular transport [34,35], nuclear pore formation [20], viral interactions [36,37,38], tumor suppression [39], and transcription [10,27,29,40]. LC8 partner proteins share a short (eight amino acid) linear recognition motif that mediates binding to LC8 [5,15,37]. The binding motif allows some variation; however, it is typically anchored by a threonine-glutamine-threonine (TQT) sequence (Figure 1B) [28]. Although it is common for LC8 partners to contain multiple LC8 binding motifs, one unique example is its own transcription factor, ASCIZ (ATMIN-Substrate Chk-Interacting Zn^2+^ finger) [40,41], which contains an astonishing eleven experimentally verified LC8 recognition motifs within its human homolog [10].

Importantly, in vivo and biophysical studies have characterized ASCIZ as a transcription factor and concentration regulator of LC8 [27,42,43]. ASCIZ is thought to act as a concentration sensor that fine-tunes LC8 transcription by interacting with LC8 via a dynamic ensemble of unsaturated bound complexes [10]. Unlike Nup159 (containing five LC8 binding sites) from yeast, which forms rigid stacked complexes, ASCIZ instead forms heterogeneous complexes as visualized by negative-stain EM analysis [10]. Drosophila (with seven LC8 sites) and human ASCIZ studies show that ASCIZ/LC8 interactions display both positive and negative cooperativity to enable this heterogeneous complexation. Such heterogeneity may be due to the disordered linkers between LC8 binding sites in ASCIZ that vary considerably in length (Figure 2). LC8Pred [15] predicts sixteen LC8 binding sites within human ASCIZ (five more than have been experimentally verified [10]). As shown in Figure 2, these binding sites can be roughly grouped into two LC8 binding domains (LBDs), with a few additional sites flanked by extensive linker regions. This trend holds true across all investigated homologs, even that from Drosophila, which contains the shortest linker between adjacent LBDs at thirty amino acids in length. This conservation of mixed long and short intra-motif spacing suggests a functional purpose to promote dynamic complexation and enable LC8 concentration sensing.

A multivalent subdomain of Drosophila ASCIZ, QT2–4, which contains the second, third, and fourth sequential LC8 binding sites, serves as a model system to probe both the highest variety in intra-site linker length and highest variability in LC8 motif strength within dASCIZ (Figure 2). Unique among experimentally verified LC8 motifs, Drosophila ASCIZ possesses an LC8 binding motif containing a TMT (QT3) rather than the canonical TQT anchor (Figure 2). Our recent studies utilizing QT2–4 provided the first evidence of in-register binding during LC8/IDP complex assembly and suggested that linker length contributes to modulating the flexibility and LC8 occupancy in multivalent LC8/IDP complexes in general [29]. The work presented here utilizes single- and double-site variants of QT2–4 to investigate the interplay of linker length and motif specificity in the regulation of dynamic, multivalent LC8 complexes. Experiments comparing the biophysical analysis of QT2–4 to QT4–6, which contains mid-length linkers, confirmed the conclusions from the variants’ study.

## 2. Materials and Methods

### 2.1. Cloning, Protein Expression, and Purification

Cloning of Drosophila ASCIZ QT2–4 (residues 271–341) with various mutations of recognition motifs was performed using either the QuikChange Lightening mutagenesis kit (Agilent) or the Q5 site-directed mutagenesis kit (New England Biolabs). The resulting constructs verified by sequencing are QT2, QT3, QT4, QT2,3, QT2,4, and QT3,4, where the number(s) following ‘QT’ indicate which LC8 recognition motif(s) remain and have not been mutated to AAA. Drosophila LC8 and ASCIZ proteins were expressed and purified according to previously published protocols [10,29]. Briefly, constructs were expressed in frame with a hexahistidine tag, Protein A solubility tag (for ASCIZ constructs), and a cleavage site for the tobacco etch virus (TEV) enzyme. All constructs were transformed into *Escherichia coli* Rosetta (DE3) cells (Merck KGaA, Darmstadt, Germany) and expressed at 37 °C in LB or ZYM-5052 autoinduction media. Recombinant protein expression was induced with 0.4 mM IPTG (for LB cultures) and growth continued at 26 °C for 16 h. Cells were harvested and purified under either regular (LC8) or denaturing (ASCIZ constructs) conditions using the TALON His-Tag purification protocol (Clontech). The solubility tag and/or hexahistadine tag were cleaved by TEV protease and the proteins were further purified using strong anion exchange chromatography (Bio-Rad, Hercules, California) followed by gel filtration on a SuperdexTM 75 gel filtration column (GE Health). Purity was assessed by SDS-polyacrylamide gels.

### 2.2. Isothermal Titration Calorimetry

Binding thermodynamics of the QT/LC8 interactions were obtained with a MicroCal VP-ITC microcalorimeter (Malvern Panalytical, Malvern, UK). All experiments were obtained at 25 °C and with protein samples in a buffer composed of 50 mM sodium phosphate, 50 mM sodium chloride, 1 mM sodium azide, 5 mM β-mercaptoethanol, pH 7.5. Each experiment was started with a 2 µL injection, followed by 27 to 33 injections of 10 µL. Experiments were conducted with QT variants in the sample cell at concentrations ranging from 20–50 μM and LC8 in the syringe at concentrations ranging from 400–500 μM. Experiments for the single site constructs resulted in calculated Brandt parameter values (c values) of 5.4, 1.4, and 3.2 for QT2, QT3, and QT4, respectively, indicating that the thermodynamic parameters for each interaction are almost out of an acceptable range for reliability. The data were processed using Origin 7.0 and fit to a simple, single set of sites binding model; however, these systems are more complicated because LC8 is a dimer binding two IDP chains. Data for the double site variants were also fit using the sequential binding sites and two sets of sites models to address the failure of the single set of sites model in satisfactorily representing the stoichiometry of binding. The reported data are from two independent experiments. In all cases, the data were reproducible. The reported concentrations are expected to have a 5–10% uncertainty in protein concentrations that were determined by absorbance measurement at 280 nm.

### 2.3. Size Exclusion Chromatography Multiangle Light Scattering

SEC-MALS was performed using a GE Healthcare AKTA FPLC with a Wyatt Technology DAWN multiple-angle light scattering and Optilab refractive index system. Experiments were performed on a GE life sciences Superdex200 10/300 GL column at room temperature equilibrated with 50 mM sodium phosphate, 0.4 M NaCl, 1 mM NaN_3_, 5 mM β-mercaptoethanol, pH 7.5 buffer at a flow rate of 0.75 mL/min. QT2–4 and QT4–6 were both prepared at 90 µM and mixed with LC8 at 300 µM resulting in a 1:3.3 ratio. Data were analyzed with Wyatt Technology ASTRA software package, version 8.

### 2.4. Analytical Ultracentrifugation

SV-AUC was performed using a Beckman Coulter Optima XL-A analytical ultracentrifuge equipped with absorbance optics. LC8 was mixed with each double site variant at ratios of 1:1, 1:2, 1:3, 1:4, 1:5, and 1:6 (molar ratio of QT:LC8). Solutions were prepared with 60 µM QT construct protein concentration. Buffer conditions for SV-AUC analysis were 20 mM Tris-HCl, 50 mM NaCl, 5 mM Tris(2-carboxyethyl)phosphine, 1 mM sodium azide, pH 7.5. The complexes were loaded into standard, 12-mm pathlength, two-channel sectored centerpieces and centrifuged at 42,000 rpm and 20 °C. A total of 300 scans were acquired with no interscan delay. The wavelength used to measure each set of experiments was chosen such that the absorbance of the sample at the given wavelength, between 280 and 302 nm, was in the 0.6–1.1 range. The data were fit to a c(S) distribution using the software SEDFIT [44]. Buffer density was calculated to be 1.0009 g/mL using Sednterp [45].

### 2.5. Native Electrospray Ionization Mass Spectrometry (Native ESI-MS)

All native mass spectra were acquired as previously described using a Waters Synapt G2-Si mass spectrometer equipped with a nanoelectrospray ionization source [29]. The instrumental settings used were as follows: source at ambient temperature, sample cone collision energy of 25 V, trap collision energy of 25 V, transfer collision energy of 5 V, and trap gas flow rate of 7–7.5 mL/min. Spectra shown represent the summation of data scans acquired over a period of 5 min. A native mass spectrum was acquired for each individual protein sample at 25 µM and used to determine accurate monomer masses. Complexes were formed by mixing LC8 with each QT2–4 mutant at a 2:1 LC8:QT molar ratio to achieve a final total protein concentration of 25 µM. After allowing complex formation to occur overnight at 4 °C, native mass spectra were acquired for each 25 µM sample, as well as for a dilution series of each at total protein concentrations of 15 µM, 10 µM, 5 µM, 1 µM, and 500 nM in 200 mM ammonium acetate at pH 7.4. After peaks in the native mass spectra were assigned, the areas of each peak were integrated with IGOR Pro 9. The summed area of each species’ various charge state peaks was used to determine relative abundances, which were then normalized to the LC8 dimer abundance for each spectrum.

## 3. Results

### 3.1. Interactions of QT2–4 Single Site Variants with LC8

We created three variants (QT2, QT3, and QT4) in which two out of three native LC8 recognition motifs in the QT2–4 construct were abolished by replacing the three TQT anchor residues with AAA so that each binding site could be studied individually while maintaining the context of the longer, disordered chain (Figure 2D). ITC at 25 °C was used to characterize the thermodynamics of these variants’ interaction with LC8. All single-site variant isotherms were fit using Origin’s “single set of sites” (SSS) model. As shown in Appendix A, ITC experiments of QT2 (Figure 3A) and QT4 (Figure 3C) with LC8 yield modest dissociation constants (K_d_) of 9.3 µM and 15 µM, respectively. The similarity of these dissociation constants is expected as the QT2 and QT4 LC8 binding sites share the canonical TQT motif anchor, and the slight affinity preference for the QT2 site supports previous results that indicate this site as the first to bind in the context of QT2–4 [29]. As expected, the interaction of QT3 with LC8 (Figure 3B) yields a much weaker binding affinity (K_d_ of 36 µM), supporting previous data on short peptides [10]. Interestingly, the ΔH and TΔS values for QT4 (−16.1 and −9.5 kcal/mol) are significantly different from those of QT2 (−10.7 and −3.9 kcal/mol) despite both containing the TQT anchoring motif. This suggests that the composition of the motif outside of the TQT anchor and/or the distance the binding site lies from the closest terminus, 9 versus 15 residues for QT4 and QT2, respectively, impact the thermodynamics of LC8 binding. For all three variants, the ΔG values are between −6 and −7 kcal/mol (Appendix A). It is worth noting that these fits ignore the context of LC8-driven duplex formation and only represent a model in which the variant is already duplexed.

### 3.2. Interactions of QT2–4 Double Site Variants with LC8

ITC experiments measuring interactions of the double site QT2–4 variants (QT2,3, QT3,4, and QT2,4) (Figure 2D) with LC8 illustrate how pairs of LC8 binding motifs interact to stabilize the duplex formed (Figure 3D–F). Since all three isotherms display a single binding step, we first modeled the binding events using SSS (Appendix A). These fits produce results that indicate that the overall binding of each protein is improved compared to the single-site variants, with lower dissociation constants and more negative free energy values. However, the N values associated with these fits are a poor representation of the reality of the complex formed.

The failure of SSS to accurately fit the N expected from these isotherms, particularly the QT2,4 and QT3,4 proteins (even the intact construct, QT2–4, although to a lesser degree) indicates that the shape of the isotherm contains multiple, convoluted sigmoidal curves. This raises the possibility that the sites in each of the double-site variants might be interacting with LC8 completely independently from one another and that the apparent increase in binding strength of each protein could simply be due to the higher concentration of LC8 binding sites in the double-site variants than exist in the single-site variants. To investigate this, we used the thermodynamic values obtained from the isotherms of the single-site variants to simulate the expected isotherms if the two sites involved in each double-site variant interact with LC8 completely independently of one another (Figure 3G–I). None of the simulated isotherms match the experimental isotherms of the double-site variants; rather, each simulated isotherm indicates weaker binding than is seen by experiment, indicating that for each of the variants, the two intact sites bind cooperatively. Due to the low c-values of the motifs in the simulated isotherms at the conditions used for the experimental isotherms at ~2.0, 1.3, and 0.56, a high degree of expected uncertainties precludes more detailed analysis.

After confirming that the isotherms of the double-site variants are not representative of completely independent LC8 binding sites, we then fit the isotherms using Origin’s “sequential binding sites” (SBS) and “two sets of sites” (TSS) models (Appendix A). SBS represents the system using a given number of binding sites that always bind to the partner in the same order. For our system, the QT2,3 protein is fit as if site 2 always binds before site 3. This assumption is reasonable because the K_d_ values of each site differ by a factor of at least 5 [46]. SBS is an imperfect fit for this system, for the same reasons previously mentioned for the fit of SSS to the single-site variants, but additionally, because concentrations may vary from measurement by up to 10% and true K_d_ values may not differ enough that the binding can be realistically expected to follow a strict progression. However, due to the presence of two disparate sites on these proteins, SBS is more appropriate than SSS and produces values that can be roughly compared to one another. TSS does not assume a binding order, and it is able to slightly adjust for concentration errors by varying the N value associated with each binding site. However, the model still must assume the QT2–4 variants are already duplexed. Additionally, because this model utilizes and reports so many parameters, the fits inherently possess higher error in each value than for the other fitting methods, as all of the terms will co-vary.

While it is true that the imperfection of the suitability of the models to our system (as illustrated in Appendix A) means that we cannot compare the precise values of the fits, we can compare the relative magnitude of the values in question. The resultant thermodynamic values map nicely to the expectation that binding affinity is increased for each double-site variant above what would be expected from independent sites. In both models, in 5 of the 6 sites in the double-site variants, the K_d_ is reduced compared to the same binding site in its respective single-site variant, ranging from a factor of 2 up to a factor of ~100 times improvement. The one site that breaks this pattern is QT3 in QT3,4, which exhibits no change in affinity. Subtle in the SBS fits, but made much more obvious in the fits to TSS, is the fact that these isotherms contain relatively little information about the second binding site in each double-site variant. While this is especially true for QT3,4 (with SBS K_d_ = 40 ± 9 µM and TSS K_d_ = 25.7 ± 13.5 µM), the large error values for N and ΔH in each weaker binding site in the TSS fits are indicative thereof. The N value of 0.2 ± 3.5 for QT3 in QT3,4 indicates incomplete binding, the N value of 0.6 ± 0.6 for QT4 in QT2,4 may indicate incomplete binding or a concentration adjustment, and the other four N values likely are only different from 1 to adjust for concentration errors. Strikingly, the K_d_ values are mostly consistent between SBS and TSS, with the one obvious deviation from this being QT4 in QT2,4. SBS indicates that the context of being coupled with QT2 increases the affinity of QT4 by 5-fold, whereas TSS indicates that it is barely stabilized at all by this context. According to SBS, K_d_ stabilization trends with the identity of the anchoring motif, in which stronger TQT motifs are strongly stabilized in the context of double-site variants, whereas the weaker TMT anchored motif is stabilized weakly or not at all in this context. TSS trends moreso with the linker length, in which a long linker stabilizes the stronger binding site greatly (10–20 fold) and the weaker binding site modestly (1.3–1.5 fold), while a short linker results in stabilizing the stronger site slightly less (~7 fold) and the weaker site slightly more so (~2 fold), for a more balanced interaction. While neither SBS nor TSS adequately models the double-site variants, analyzing the results of both provides the most complete picture of the binding events in this system as attainable by ITC.

As a reminder, SBS is a reasonable approach if the K_d_ values involved are different by at least 5-fold. The K_d_ values for LC8 binding to QT2–4 at QT2 and QT4 are not different enough for the model assumptions to be reasonable; however, because QT2–4 contains three binding sites, the other models cannot assess this system at all. Thus, the values derived must be considered cautiously. With this in mind, the values fit to the QT2–4 isotherm (reanalyzed for the purpose of this discussion [10]) may reveal interesting insights into this complex. For instance, the addition of a third binding site reduces the affinity of each binding site in relation to the double-site variants (barring QT3 in QT3,4 and possibly QT4 in QT2,4). Because this effect is seen from each of the double-site variants in comparison to the intact three-site protein indicates that multiple factors play into this property. Comparison of both variants with long linkers between intact binding sites (QT2,4 and QT3,4) to the QT2–4 construct suggests that the inclusion of a weak motif between two relatively strong motifs results in steric pressure on the duplex when the third LC8 attempts to intercalate between the other two sites that are already bound. Contrastingly, when QT2,3 is compared to QT2–4, the addition of site 4 introduces a long linker into the context of binding, and this linker results in a reduction of the affinity of both QT2 and QT3. This suggests that the long linker also contributes to the negative cooperativity and additional heterogeneity of LC8 binding to ASCIZ. The evidence of multiple sources of heterogeneity is particularly interesting when considering the role ASCIZ plays in sensing and regulating the cellular concentration of LC8. It follows that the various contributors to allostery in ASCIZ binding LC8 allow ASCIZ to experience a wider variety of bound states in response to a broad range of LC8 concentrations, an important feature for a quality cellular concentration sensor.

### 3.3. Complex Formation Monitored by Sedimentation Velocity Analytical Ultracentrifugation (SV-AUC)

To further investigate the heterogeneity of complexes formed between the QT2–4 double-site variants and LC8, we used SV-AUC to track QT/LC8 complex assembly. For these experiments, peaks indicate the presence of LC8, whether alone or in complex, because the extinction coefficient of QT2–4 at 280 nm is too small to be measured. SV-AUC analysis of the double site constructs in complex with LC8 show that the proteins are in dynamic equilibrium at molar ratios of QT:LC8 up to 1:5 for QT3,4 (Figure 4A), 1:3 for QT2,4 (Figure 4B), and above 1:6 for QT2,3 (Figure 4C) and that the complexes formed by each variant at each LC8 ratio vary from one another in their sedimentation coefficients. Complexes formed with QT3,4 have sedimentation coefficients of ~2.5, ~3.8, ~3.9, ~4.1, and ~4.25 S at QT:LC8 ratios of 1:1, 1:2, 1:3, 1:4, and 1:5, respectively (Figure 4A). Complexes formed with QT2,4 have sedimentation coefficients of ~2.8, ~3.9, and ~4.1 S at ratios of 1:1, 1:2, and 1:3 (Figure 4B). Finally, complexes formed with QT2,3 have sedimentation coefficients that increase approximately linearly from ~3.3 to ~4.3 S along the measured ratios (Figure 4C) and, based on the trend, may continue to grow at higher ratios.

The sizeable shift in sedimentation coefficient for QT3,4 and QT2,4 complexes between the ratios of 1:1 and 1:2 is indicative of the convolution of complex with free LC8. Knowing that free LC8 has a peak at S ≈ 2, the complex peaks in QT3,4 and QT2,4 are likely close to S = 3.5. QT2,3 at 1:1, however, has almost no free LC8 peak, which indicates nearly complete binding of the available LC8. These values are consistent with the shifts seen for each of the subsequent titration points, which indicate the equilibrium of the mixture moving toward a fully bound 2:1 (LC8:QT) complex. While QT2,3 is the most efficient at binding LC8 early in the titration, QT2,4 plateaus at the earliest titration point (1:3) and QT3,4 plateaus at a 1:5 ratio at a slightly higher sedimentation coefficient. The sedimentation coefficient of the complex peak for QT2,3/LC8 at the 1:2 ratio is lower than those seen for QT3,4 and QT2,4; this can be explained by the close proximity of QT2 and QT3 to the N-terminus which leaves a long, unbound tail which increases the frictional ratio of the complex (Figure 4C). The continuing increasing value of S in the QT2,3 AUC titration at high concentrations of LC8 may suggest an alternative binding mode that begins to occur at high concentrations of LC8, such as an offset structure that allows three LC8 dimers to bind a pair of QT2,3 chains. This structure, while perhaps not intuitive, is favored as per Le Châtelier’s principle, in which a greater number of partially bound LC8 dimers becomes more favorable than a fewer number of fully bound LC8 dimers. However, SV-AUC cannot directly inform on the stoichiometry of complexes formed.

### 3.4. Complex Formation Monitored by Native ESI-MS and EM

Using native electrospray ionization (ESI)-MS, measurements of individual protein mixtures with LC8 allow for the characterization of complex stoichiometries. Similar experiments were used previously to study the complex formation of QT2–4 with LC8 [29]. Accurate mass determination for each protein matches closely with the expected masses of each sequence (Appendix A). Upon conducting dilution series, QT3,4, QT2,4, and QT2,3 each remain as monomeric chains while LC8 is overwhelmingly dimeric in solution.

Further native mass spectra acquired for 2:1 mixtures of LC8 with double-site variants indicated the same complex stoichiometries exist for each variant (Figure 5, Appendix A). The four detected complexes correspond to species with variant:LC8 ratios of 1:2, 1:4, 2:4, and 2:6. These results mimic those determined for wildtype QT2–4/LC8 complexes, as both the expected “in-register” complex (2:4) and an “off-register” complex (2:6) are present. Note that QT2,3 exhibits the greatest population of 2:6 complex consistent with the hypothesis that it forms off-register complexes more readily than the other two variants. Of note, in-register complexes are always detected in greater abundance than off-register complexes for all double-site constructs, but the persistence of in- and off-register complexes at low concentrations indicates they are each naturally occurring rather than spurious [29] (Figure 5).

While QT3,4, QT2,4, and QT2,3 all form the same set of complexes with LC8, the detected abundances vary between the systems (Figure 5B). Of the four complexes detected, the 2:4 in-register complex (one QT duplex, two LC8 dimers) is the most abundant species formed by QT2,3 at nearly all concentrations studied, indicating high cooperativity between sites QT2 and QT3. In contrast, both QT2,4 and QT3,4 form the intermediate 1:2 complex (one QT chain, one LC8 dimer) as the most abundant species detected across all concentrations. These results indicate that LC8 binding to QT2,3 is more cooperative than QT2,4 or QT3,4. This is consistent with ITC and SV-AUC results presented above.

The complex species identified with native ESI-MS also provide evidence for a potential mechanism of complex formation. First, an LC8 dimer binds to a single chain, forming the 1:2 species. This is followed either by binding a second LC8 dimer, resulting in a 1:4 complex, or by binding to another 1:2 species and rearrangement to a symmetric 2:4 species. If the former path occurs, a second protein strand is subsequently recruited to form the expected in-register 2:4 complex. Misalignment of the second strand by either pathway would allow a third LC8 dimer to bind, resulting in an off-register 2:6 complex (one QT duplex, three LC8 dimers). Figure 5C depicts the proposed mechanism of assembly and ensembles of complexes formed by each variant.

Electron microscopy (EM) images (Figure 5D) collected of mixtures of LC8 with QT2–4 match the conclusions made from MS data. Relative proportions of strings of two, three, and four LC8 dimers observed by EM are plotted and indicate a large excess of species with two LC8 dimers attached by QT2–4 proteins and small amounts of species with three or four LC8 dimers. From the AUC and MS results, it seems reasonable to conclude that the majority of the species with two LC8 dimers are bound at QT2 and QT3.

### 3.5. Comparison of the Complex Heterogeneity of LC8 Bound to QT2–4 versus QT4–6

To further test the conclusions gleaned from the variants of the QT2–4 construct, we compared the intact QT2–4 construct to a different ASCIZ subdomain, QT4–6 (Figure 2C). Previous ITC [10] has shown QT4–6 to bind LC8 more tightly with an N of 3 and K_d_ of 1.0 µM, compared to an N of 2.7 and K_d_ of 4.1 µM for QT2–4 (Appendix A). To further this comparison, we characterized the complexation of each construct with LC8 by AUC titration and Size Exclusion Chromatography MultiAngle Light Scattering (SEC-MALS). AUC titrations illustrate that the QT2–4 complex (Figure 6A) forms later in the titration than the QT4–6 complex (Figure 6B). As described previously for QT2,4 and QT3,4 AUC, the peak seen in QT2–4 at 1:1, with sedimentation coefficient 3.0, is evidence of a convolution of lower occupancy complex with free LC8, whereas QT4–6 traces do not exhibit free LC8 until the 1:3 ratio. Furthermore, the LC8 peaks in QT2–4 do not line up with the LC8 alone trace at any titration point, while the QT4–6 LC8 free peak lines up consistently, indicating that QT2–4 contains a small population of low occupied complex even at high titrations while the same is not true for QT4–6. Lastly, the QT4–6 titration shows saturation by the 1:4 ratio, while that is not observed for QT2–4, evident by the position of the LC8 free peak. These together indicate that QT4–6 binds LC8 highly cooperatively and uniformly but that QT2–4 binds LC8 much more heterogeneously.

SEC-MALS analysis of the QT2–4 construct complexed with LC8 (Figure 6C) indicates that the major species involves duplexed QT2–4 linked by one LC8 dimer (QT_2_LC8_2_). However, peak 1, which contains free LC8, does not align with LC8 when run alone, indicating that a significant amount of complex disassociated on the column and that the complex upon injection may have been the QT_2_LC8_4_ complex. Contrastingly, the major species in the complex of QT4–6 with LC8 (Figure 6D) is a mixture of a QT4–6 duplex bridged by two or three LC8 dimers. The width of the LC8 free peak in QT4–6 indicates a minor population of complex dissociated on the column. These results are again consistent with the conclusion that QT2–4 forms a more heterogeneous complex than QT4–6.

Analysis of the binding motifs present in each construct by ITC of peptides has been conducted previously [10] (Figure 6E). K_d_ values indicate that the difference observed between these two constructs cannot be attributed simply to a better set of binding motifs in QT4–6 than is present in QT2–4; in fact, the opposite might be claimed wherein the QT2 motif is much more favorable for LC8 binding than any of the other motifs involved in either construct. Thus, if the incomplete binding and heterogeneous behavior of the QT2–4 construct cannot be attributed to motif stability and specificity, then it must be attributed to the varying linker lengths that are found in that construct. This also indicates this region as an origin of the dynamic ensemble that is observed in dASCIZ and its homologs in general.

## 4. Discussion

LC8 commonly forms duplex scaffold assemblies with its many multivalent IDP partners [13,14,15], and cases in which the IDP ligand contains multiple binding sites for LC8 continue to emerge. However, the contribution of multivalency to complex stability and heterogeneity is not fully understood. Variability in both motif specificity and linker lengths between motifs are well represented in Drosophila ASCIZ, especially within the QT2–4 subdomain. Serving as a model system, this construct contains both the shortest and longest linkers between LC8 binding sites as well as an uncommon TMT LC8 anchor motif. A recent study utilizing QT2–4 provided the first confirmation of in-register binding during LC8/IDP complex assembly and showcased the role that linker length plays in modulating the flexibility of such complexes [29]. The work presented here expands on these results by investigating how the interplay of linker length and motif specificity regulate the compositional heterogeneity of dynamic, multivalent LC8 duplexes. Additionally, a comparison of QT2–4 to QT4–6, another construct from Drosophila ASCIZ, further illustrates the regulatory role played by short and long linkers.

### 4.1. Two LC8 Binding Sites Are Cooperative, but a Third Site Is Negatively Cooperative

ITC experiments of single site variants provide motif-specific binding affinities in the context of the QT2–4 disordered chain for QT2, QT3, and QT4 (Figure 3A–C). QT2 and QT4 show similar binding affinities (9.3 and 15 µM, respectively), while QT3 is considerably weaker (36 µM). This is expected because QT3 contains a TMT anchor that is weaker than the TQT anchors found in QT2 and QT4. The slight favorability for QT2 is consistent with prior evidence which indicates QT2 as the initial site of LC8 binding within QT2–4 [29]. With double site variants, ITC indicates variability in LC8 binding, but always with positive cooperativity (Figure 3D–F). When a third binding site is introduced, affinity decreases for all binding sites. In the QT2–4 system, we are unable to distinguish if location or motif specificity plays a larger role in imparting negative cooperativity in a triple-site, multivalent IDP compared to a double-site. However, because of the sizeable decrease in affinity shown here, we conclude that both properties are likely at play. In fact, motif specificity seems to be tuned by multivalency: stronger motifs benefit more from a second binding site but are also hindered more so by a third binding site. The discussion of linker length effects, however, is more intricate.

### 4.2. QT2,3 Forms Stable Complexes with LC8 More Readily than Do QT3,4 and QT2,4

Though the double-site variants have the same number of binding sites for LC8 dimers, it is clear they differ in their complex assembly. Although K_d_ values calculated by all three fitting methods (SSS, SBS, and TSS) indicate that QT2,4 forms the most cooperative complex in comparison to the single site variants, the N values calculated by SSS and TSS both indicate that QT2,3 binds LC8 more completely than the other two constructs. This is further substantiated by the AUC and native MS results in which no excess LC8 is present at low titration points of QT2,3 AUC and where MS shows a greater proportion of QT2,3 is complexed in a 2:4 stoichiometry (QT:LC8) than is seen for the other variants. We attribute this degree of cooperativity to the very short linker in QT2,3 which is 3 residues long, compared to the longer linkers, 30 and 41 residues in length. This indicates homogeneous binding to QT2,3 at low LC8:QT ratios compared to QT2,4 and QT3,4, which both bind heterogeneously at these ratios.

However, at higher ratios of LC8:QT, the aforementioned trends persist and imply heterogeneous binding of LC8 to QT2,3 in these concentration regimes. The AUC results for QT2,3 at titration points of 1:4, 1:5, and 1:6 exhibit a continued increase in sedimentation coefficient further than expected. Combined with the results from MS that show that QT2,3 forms more of the 2:6 (QT:LC8) complex than the other two variants, this suggests that this complex is becoming more populated in the higher titration points of the AUC which leads to an increased sedimentation coefficient. We hypothesize that at high LC8 concentrations, the QT2,3 complex assembles as shown in Figure 5C, in which the chains slide into an offset registration and two of the LC8 are only half bound. The extremely short linker may enable this complex to be stabilized by lateral contacts between adjacent LC8 dimers, perhaps via van der Waals interactions. This mode of complexation can be further explained with the application of Le Châtelier’s Principle to this system in which higher ratios of LC8 put pressure on the equilibrium to favor a higher population of partially bound LC8 over a small population of fully bound LC8 accompanied by a large population of unbound LC8. It is unsurprising that no evidence of daisy-chaining is seen in our experiments because steric hindrance would likely preclude any chain from binding to the free side of the half-bound LC8 dimers, let alone the entropic penalty of binding a stiff chain of LC8 dimers. This effect is not seen in the variants with longer linkers because they lose the lateral contacts between LC8 dimers and result in long, extended structures (Figure 5C).

### 4.3. Linker Length Is More Important than Motif Specificity for Determining Heterogeneity of LC8 Binding

While we have discussed that the very short linker in QT2,3 induces heterogeneous binding at higher LC8 concentrations, it is also true that the long linker present in the other variants induces heterogeneous binding, especially at lower concentrations of LC8. ITC (Figure 3) and AUC (Figure 4) both indicate incomplete binding of LC8 through the low N values fit by SSS and TSS to ITC and the presence of free LC8 at low titration points by ITC. Moreover, although the double-site variants each form the same four complex species in solution as determined by native MS, at stoichiometries of 1:2, 1:4, 2:4, and 2:6 (QT:LC8), they vary significantly in their detected abundance (Appendix A, Figure 5). Single chain complexes (1:2 and 1:4) are more abundant for the variants with long linkers (QT3,4 and QT2,4) than for QT2,3. For QT2,4 and QT3,4, these single-chain complexes are also more abundant than the duplex species, even at the highest concentration tested (Figure 5). This indicates that increasing the linker length between LC8 binding sites disrupts duplex formation of IDP multivalent complexes. Similarly, EM results indicate an overwhelming proportion of species with only two LC8 dimers bound to QT2–4 protein strands, presumably bound to the QT2 and QT3 binding sites. Additionally, a comparison of SEC-MALS (Figure 6) of QT2–4, which contains the lengthy linker, and QT4–6, which contains moderate-length linkers, shows that QT4–6 assembles as a dimer with either two or three LC8 dimers bound at the same conditions in which QT2–4 dimers only bind to one LC8 dimer. Of note, the QT2–4 complex peak is broad and the LC8 peak is shifted, indicating a more heterogeneous mixture of complexes and a more dynamic assembly than is seen with the QT4–6 construct. Interestingly, these differences cannot be explained by differences between motif specificities involved in each construct because a comparison of the motifs between QT2–4 and QT4–6 indicates similar binding strengths (Figure 6E). It is worth noting, however, that motif specificity remains important to complex formation. The weak-binding TMT motif in QT3,4 causes a lower overall LC8 binding affinity by ITC compared to QT2,4, which contains a similar length linker (Figure 3E, Appendix A) and requires a greater ratio of LC8 to reach saturation by AUC (Figure 4). Additionally, the weak TMT motif is likely part of what enables the offset structure proposed for QT2,3 through dynamic binding to LC8. Comparison of the double-site variants and the intact constructs QT2–4 and QT4–6 suggests a “Goldilocks” zone for linker length in regard to non-heterogeneous binding wherein the short 3 residue linker and the long 30 and 41 residue linkers result in heterogeneous binding, but the mid-length 6 and 9 residue linkers are not associated with heterogeneous binding. Together, these results highlight the importance of both linker length and motif specificity and determine their interplay as a regulation mechanism for IDP/LC8 multivalent complex assembly.

We have shown that short linkers and long linkers both contribute to heterogeneity, while 6 and 9 amino acid linkers result in homogeneous complexation. However, we have not established the barrier between a “mid-length” linker and a long linker. In dASCIZ, between QT6 and QT7, there is a 12 amino acid linker and between QT1 and QT2, there is a 16 amino acid linker. Previous research shows that the QT4–7 and QT1–3 constructs have binding affinities that fall between those of QT4–6 and 2–4 [10]. Of note, this means that QT1–3 has a lower affinity than QT2,3 and that QT4–7 has a lower affinity than QT4–6. While these constructs are not identical in their contextual residues, the differences do suggest that the addition of QT1 and QT7 both result in poorer binding systems. This may indicate that 12 and 16 amino acid linkers are already long enough that they begin to induce heterogeneity in LC8/ASCIZ complexation. Further study of these constructs and of double site variants of these and QT4–6 (QT4,6 contains a 23 amino acid linker) will help to elucidate the barrier between mid-length linkers, which lead to homogeneous binding, and long linkers, which induce heterogeneity at low LC8 concentrations.

Interestingly, human ASCIZ contains a run of 4 LC8 binding sites (Figure 2) with linkers of 1, 6, and 24 residues, respectively, and quite intriguingly, the second of these binding sites has a TMT anchor sequence. While we by no means believe this to be the only source of heterogeneity in LC8 binding to hASCIZ, we hypothesize that it is a strong contributor to the formation of the dynamic complex that has been described for this system [10].

## 5. Conclusions

Herein we show that binding of LC8 to multivalent QT2–4 variants is complex and governed more strongly by the length of disordered linkers between LC8 binding sites than by LC8 motif specificity. Cooperativity between multivalent sites is positive for double-site variants but negative for the three-site construct QT2–4. Additionally, the multivalent constructs with short linkers between sites resulted in stable saturated LC8/IDP assemblies that are readily formed in solution compared to constructs with longer linkers that showed a greater propensity for the formation of unsaturated complexes. Comparison of constructs with similar linker lengths, but variability in motif specificity, emphasize that both properties are involved to varying degrees in regulating IDP/LC8 complex assembly. While our initial hypothesis that long linkers contribute to heterogeneous binding was validated by our findings, it is also evident from our experiments that very short linkers similarly contribute to heterogeneous LC8 binding at high concentrations, matching observations that ASCIZ/LC8 complexes are heterogeneous at all concentrations. These findings are important for understanding the behavior of the hASCIZ/LC8 complex and suggest regions that should be studied further, which may contribute to heterogeneity. In particular, the long linker between LBD1 and LBD2, as emphasized in Figure 2, but also the region between F641 and N750 containing a 1 residue linker, a mid-length linker, a long linker, and a TMT anchored LC8 binding site. Our work is also applicable to the study of other ordered hubs binding their partners and to IDPs with multiple partner binding sites, whether for one or multiple distinct partners. Partner binding will be regulated by the lengths of the disordered linkers between each site and the strength of the binding sites involved.

## Figures and Tables

**Figure 1 biomolecules-13-00404-f001:**
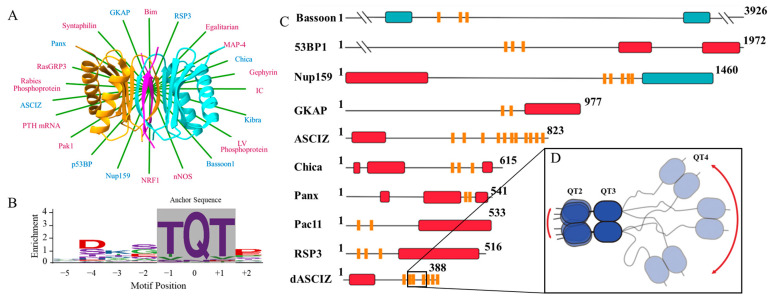
LC8 hub, binding motif, and multivalent partners. (**A**) Ribbon diagram of the LC8 dimer showing each monomer (orange and cyan) bound to disordered peptides (magenta) that adopt β-strand structure upon binding in LC8’s binding groove (Protein Data Bank code 2P2T, from *D. melanogaster*). Ribbon diagram is overlayed on a star display of a selection of LC8 partner proteins. Red font denotes monovalent partners, while blue denotes multivalent partners. (**B**) Amino acid enrichment for each position in the LC8 binding motif. The TQT anchor is boxed in gray. (**C**) Multivalent LC8 binding partners. Sequence-based predictions of order (red boxes), disorder (black lines), coiled-coil (blue boxes), and LC8 binding motifs (orange bars) are shown. PSIPRED [16] was used to predict order and disorder. Paracoil2 [17] was used to predict coiled-coils. LC8 binding motif locations are shown for Bassoon [18], 53BP1 [19], NUP159 [20], GKAP [21], ASCIZ [22], Chica [23], Panx [24], Pac11 [25], RSP3 [26], and dASCIZ [27]. Panel adapted from Clark et al. [28]. (**D**) Zoom of QT2–4 from dASCIZ showing the effect of a long linker on flexibility. Panel adapted from Reardon et al. [29].

**Figure 2 biomolecules-13-00404-f002:**
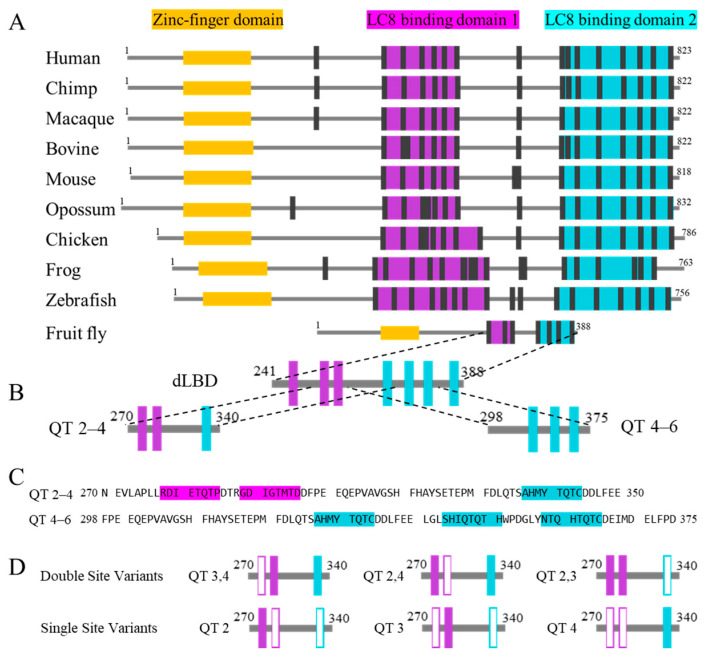
Alignment of ASCIZ homologs’ domain architecture and dASCIZ constructs. (**A**) Comparison of 10 ASCIZ homologs, aligned to show the similarity of the LBDs and the linker connecting LBD1 to LBD2. (**B**) Drosophila ASCIZ LC8 binding region and the constructs utilized in this study, including QT2–4 and QT4–6. (**C**) Sequences of QT2–4 and QT4–6. (**D**) Variants that systematically abolish either one (**top**) or two (**bottom**) LC8 recognition motifs from QT2–4 through mutation of the anchor triplet to AAA. Hollow boxes indicate sites that have been mutated. Construct nomenclature denotes the binding sites left intact in each construct.

**Figure 3 biomolecules-13-00404-f003:**
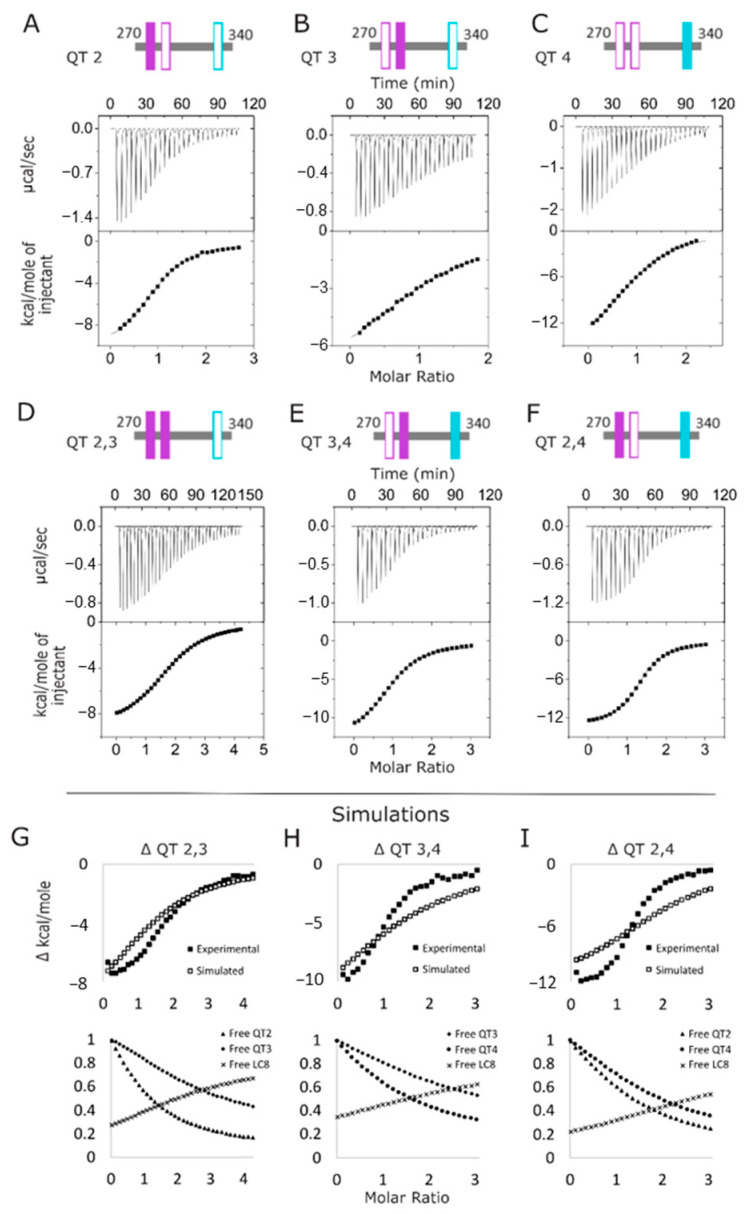
LC8-ASCIZ interactions monitored by ITC. (**A**–**F**) Representative thermograms of the titration of LC8 into QT2–4 variants corresponding to the single-site constructs QT2 (**A**), QT3 (**B**), and QT4 (**C**), and the double site constructs QT2,3 (**D**), QT3,4 (**E**), and QT2,4 (**F**). (**G**–**I**) Simulated isotherms overlaid on experimental isotherms for each of the double-site variants: QT2,3 (**G**), QT3,4 (**H**), and QT2,4 (**I**). Isotherms were simulated using ΔH and K_d_ values obtained from single-site isotherms. Fractions of free sites at each point in the simulated isotherms are shown below each, respectively.

**Figure 4 biomolecules-13-00404-f004:**
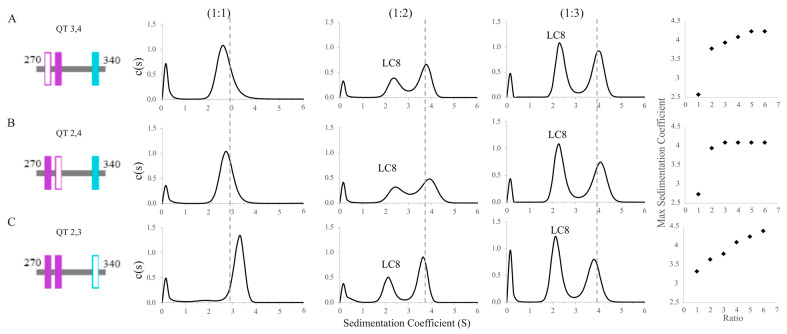
Sedimentation velocity analytical ultracentrifugation of double site ASCIZ constructs bound to LC8. SV-AUC titrations of QT3,4 (**A**), QT2,4 (**B**), and QT2,3 (**C**) with LC8 at three separate molar ratios of QT:LC8 (1:1, 1:2, 1:3) and a plot of LC8:QT ratio vs. complex sedimentation coefficient up to a ratio of 1:6. When applicable, populations corresponding to free LC8 are labeled. The dashed lines correspond to the location of the peak seen in QT2–4 at the same ratio.

**Figure 5 biomolecules-13-00404-f005:**
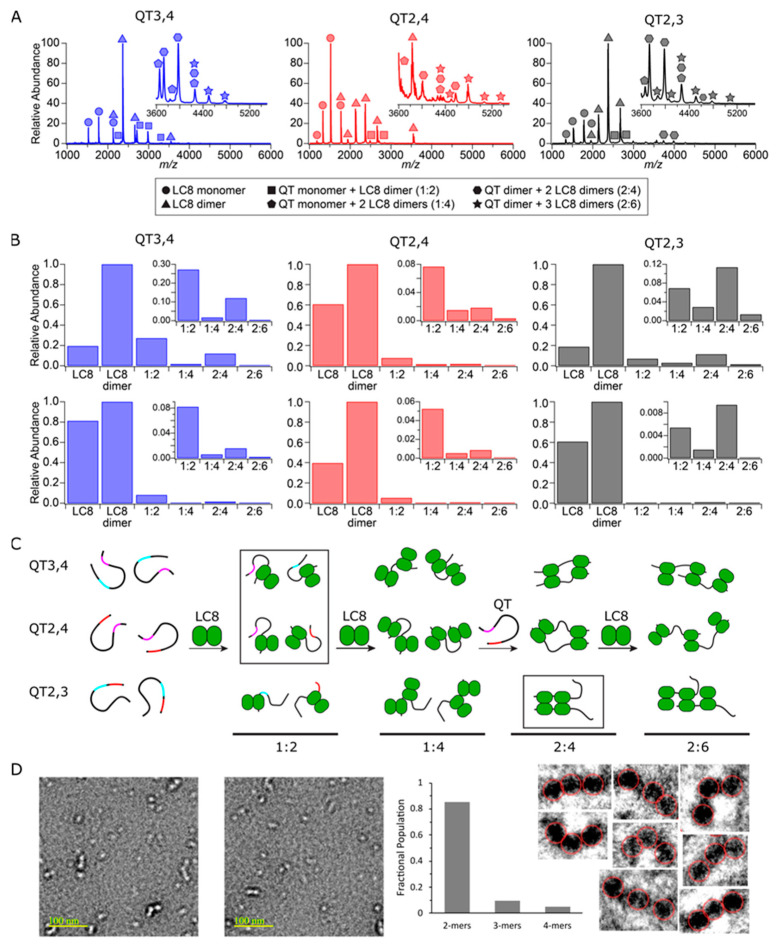
QT/LC8 complex species and abundance distributions determined by ESI-MS and EM. (**A**) Native mass spectra of double-site variants at 25 µM are shown with individual and complex species labeled. (**B**) Abundance distributions of species detected at 25 µM (**top**) and 5 µM (**bottom**) for each double-site variant are shown. (**C**) Monomeric chains of each double site variant with QT2, QT3, and QT4 sites color coded. Upon addition of LC8, 1:2, 1:4, 2:4, and 2:6 complexes form. The most abundant complex species for each QT construct is boxed. (**D**) Two representative EM images out of 50 captured of QT2–4/LC8 mixture in which bright dots in the images indicate LC8 dimers. Plotted relative populations of complexes seen in EM, and zoomed negatives of all eight 3-mers observed in the 50 images.

**Figure 6 biomolecules-13-00404-f006:**
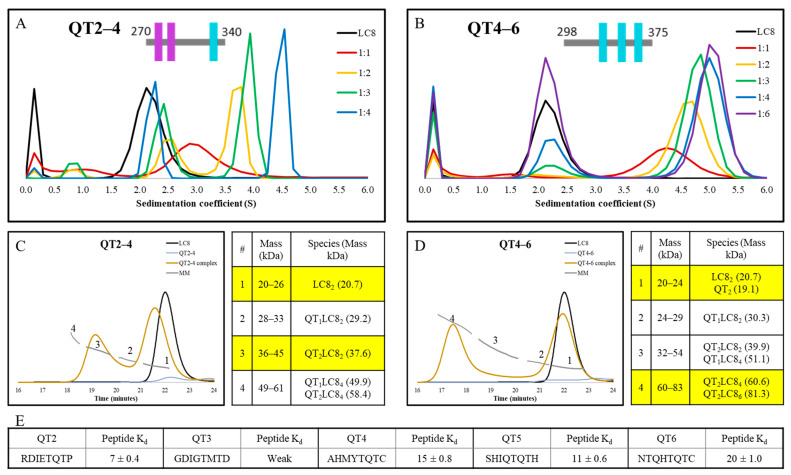
AUC, SEC-MALS, and peptide K_d_ comparisons for the constructs QT2–4 and QT4–6. AUC titration of (**A**) QT2–4 (replotted data from Reardon et al. [29]) and (**B**) QT4–6. SEC-MALS chromatogram and mass key for ranges numbered as shown on the chromatogram for (**C**) QT2–4 and (**D**) QT4–6 as single proteins and in complex with LC8. LC8 alone trace is shown plotted with both proteins for reference. Highlighted regions of the mass key emphasize the major peaks seen in the SEC-MALS traces of the mixture. (**E**) Measured K_d_ values for peptides representing the five binding sites [10] represented across the two analyzed constructs.

## Data Availability

The data presented in this study are available upon request from the corresponding author.

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
