# Peer review of "Linker Length Drives Heterogeneity of Multivalent Complexes of Hub Protein LC8 and Transcription Factor ASCIZ"

_biomolecules, 2023, doi:10.3390/biom13030404_

Round 1

Reviewer 1 Report

The manuscript by Walker et al. describes the binding of LC8, a hub protein, with regions of the ASCIZ protein that contain LC8 motifs. Multiple LC8 motifs occur in many proteins and are separated by linkers that are variable in length. Here, the authors use a region of ASCIZ (subdomain QT2-4) to assess how the linker spacing affects LC8 binding. The authors use a number of biochemical and biophysical approaches and conclude that the linker spacing is a critical parameter that controls complex heterogeneity and specificity.

Because the QT2-4 subdomain that includes three LC8 binding sites, and LC8 dimer can potentially bind two sites, this study by is very nature is very complicated. Indeed, the authors to an admirable job of tackling this study in methodologically way by using ASCIZ QT2-4 constructs with either one, two or three intact binding sites. The ITC data alone would be impossible to assess without the entire set of experiments performed. Although still difficult to assess, the authors distill the important information and discuss the limitations of their analyses. This is critical for this paper. In the end the authors glean a substantial amount of binding information with respect to the relative complexes. The use of AUC, MS, EM, and SEC-MALS together shows the power of these methods to assess individual complexes within a complex mixture. In the end, the authors provide some insight into the origin of ASCIZ/LC8 complex heterogeneity previously reported by this same group.

The main weakness (minor) in the manuscript is that the authors do not extrapolate more about how the other LC8 binding sites in ASCIZ might impact on the conclusions found in this study. Or at least indicate what might be the next big questions to address.   

Reviewer 2 Report

Suggestions for Improvements:

1. Title: many readers will not be familiar with “LC8/ASCIZ” - suggest using the term “hubprotein LC8 with transcription factor ASCIZ” in the title for further clarification

2. Abstract: 

Line 11: Which LC8 are we talking about here? Mammalian? Human?

Line 13: the term “LC8 motif” is used without explanation or definition - suggest adding a brief definition of what they are, what they do

Line 18: would not “inter-motif spacing” be a better description that “inter-motif length”?

Line 20: What does “off set structures” mean in this context? Clarify.

Line 22: the terms “dimers” and “trimers” are more commonly used.

Line22/23: the term “negative stain” is unnecessary detail here

3. Introduction

Line 39: move reference [2] before full stop. This is also an issue in Line 49, 54, 60, 69 etc. (essentially everywhere) - reference go WITHIN the sentence, not after the full stop!! Anybody who reads the scientific literature regularly should have noticed that a long time ago.

Line 75: what species is that LC8 dimer from? Need to mention that nere.

Line 88: the “dimerization engine” concept needs to be explained in more detail - what does this mean?

4. Materials and Methods

Line 146/147: Even if published previously, a brief description  of the nature of the expression system (bacterial, yeas, baculovirus, etc.) would be helpful here. Are the proteins expressed in soluble form, tagged, refolded etc?

Line 151/152: Were these buffers optimized for the binding in any way? It is relatively low salt and misses divalent ions - is that at all relevant?

Line 164: What do the authors mean by “In all cases the data were reproducible”?  To what extent? Two independent experiments are not really enough for even minimal statistical relevance.

5. Results

Nice combination of multiple approaches to study the question from all angles - the major strength of this paper!

Discussion

Very good, but I would like to see a diagram that summarizes the results and concepts in a graphical manner - this really would help readers to follow the arguments presented in the text.

I was also wondering whether a sequence alignment of ASCIZ orthologs from, for example different insect species, would show some evidence in terms of conservation of linker lengths. Have the authors looked into that?

Reviewer 3 Report

In the manuscript, Douglas R. Walker et al. focused on characterizing the LC8/ASCIZ multivalent complex. Three variants, including QT2, QT3, and QT4, have been created for this study. The results demonstrate that the heterogeneity of complexes is significantly affected by the length of the disordered linker between LC8 binding sites. Overall, the manuscript is well-written and suitable for publication in Biomolecules journal after addressing the following questions:

  1. One critical question is about the heterogeneous complex's interaction mechanism. The protein dimers can commonly interact with each other when connected with a short linker. The short linker will enhance the lateral contacts between adjacent LC8 dimers. I am not sure current data can support the proposed model in Figure 5. If higher-order complexes are formed via van der Waals interaction, why do they form regular structures (Figure 5C) other than the random aggregates? It is not convincing that the "in-resister" or "off-register" will exist. Could the authors provide any structure models of the complex?
  2. The EM images are confusing, and the scale bar is almost invisible. Please highlight the dimers, trimers, and tetramers in the images. If LC8 tends to fold homodimer, why is the fraction of the trimer higher than the value from tetramer when mixing it with QT2-4? The stick-and-ball cartoon is strange. Usually, the protein structure will change slightly during the deposition of samples on the surface, especially with flexible linkers. Could the authors perform the AFM experiments in solution and minimize the artifact during sample preparation?
  3. Is the simulated data from the isotherms of the single-site variant appropriate for comparison with the double-site variant? I also suggest providing the simulated control data of the double-site variant.

Minor issues:

  1. Please change the color of one LC8 monomer to yellow or cyan in Figure 1A.
  2. The location of the citation footnote should be consistent in the main text. Some footnotes are on the right side of the full stop.
  3. In line 289, what is the physical meaning when the authors had 0.2 ± 3.5? Does the error only come from poor fitting?

Round 2

Reviewer 3 Report

The authors have addressed all my questions. It is suitable for the publication in Biomolecules journal.